# Structure of the transcription open complex of distinct σI factors

Jie Li[1,2,3,4,5,6,10], Haonan Zhang[6,7,10], Dongyu Li[6,7], Ya-Jun Liu[1,2,3,4,5,6], Edward A. Bayer [8,9], Qiu Cui[1,2,3,4,5,6], Yingang Feng [1,2,3,4,5,6,11] ✉ & Ping Zhu [6,7,11] ✉

Bacterial σI factors of the σ70-family are widespread in Bacilli and Clostridia and are involved in the heat shock response, iron metabolism, virulence, and carbohydrate sensing. A multiplicity of σI paralogues in some cellulolytic bacteria have been shown to be responsible for the regulation of the cellulosome, a multienzyme complex that mediates efficient cellulose degradation. Here, we report two structures at 3.0 Å and 3.3 Å of two transcription open complexes formed by two σI factors, SigI1 and SigI6, respectively, from the thermophilic, cellulolytic bacterium, *Clostridium thermocellum*. These structures reveal a unique, hitherto-unknown recognition mode of bacterial transcriptional promoters, both with respect to domain organization and binding to promoter DNA. The key characteristics that determine the specificities of the σI paralogues were further revealed by comparison of the two structures. Consequently, the σI factors represent a distinct set of the σ70-family σ factors, thus highlighting the diversity of bacterial transcription.

Bacterial σ factors are critical components of RNA polymerase (RNAP) holoenzymes for initiation of transcription by specifically recognizing DNA promoter regions[1,2]. A single bacterium often contains multiple σ70 factors, which have been further classified into four groups, according to sequence conservation and domain architecture[3]. Group I includes housekeeping factors containing four conserved regions (σR1 to σR4) which are further divided into subregions[4], largely corresponding to different domains (σ1 to σ4) by structural studies[5]. Groups II to IV include alternative σ factors regulating genes for specific functions. Group IV harbors only σ2- and σ4-domains, also termed ExtraCytoplasmic Function (ECF) σ-factors[6,7], which are functionally diverse for bacterial signaling in response to various external stimuli[3]. σ2- and σ4-domains recognize promoter −10 and −35 elements,

respectively, and ECF σ-factor domains exhibit high specificity for promoter recognition, which is valuable in synthetic biology[7,8]. Structures of RNAP in complex with different groups of σ factors, including housekeeping σA (group I), σS (group II), σ28 (group III), ECF σ-factors σH, σL, and σE (group IV), and σN (family σ54), have illustrated how bacterial σ factors specifically recognize promoters and initiate transcription[9–19].

All of the known σ70 family σ factors in these structures specifically recognize the promoter −35 and −10 elements by σ4 and σ2 domains, respectively. In addition to the σ2 and σ4 domains, the group I σ factor contains σ1.1 and σ3 domains, which function in the auto-regulation of σ70 and the binding of the extended −10 region, respectively[9,10]. The group II σ factors lack σ1.1 but exhibit high sequence identity with the

[1]CAS Key Laboratory of Biofuels, Qingdao Institute of Bioenergy and Bioprocess Technology, Chinese Academy of Sciences, 266101 Qingdao, Shandong, China. [2]Shandong Provincial Key Laboratory of Synthetic Biology, Qingdao Institute of Bioenergy and Bioprocess Technology, Chinese Academy of Sciences, 266101 Qingdao, Shandong, China. [3]Shandong Engineering Laboratory of Single Cell Oil, Qingdao Institute of Bioenergy and Bioprocess Technology, Chinese Academy of Sciences, 266101 Qingdao, Shandong, China. [4]Shandong Energy Institute, 266101 Qingdao, Shandong, China. [5]Qingdao New Energy Shandong Laboratory, 266101 Qingdao, Shandong, China. [6]University of Chinese Academy of Sciences, 100049 Beijing, China. [7]National Laboratory of Biomacromolecules, CAS Center for Excellence in Biomacromolecules, Institute of Biophysics, Chinese Academy of Sciences, 100101 Beijing, China. [8]Department of Biomolecular Sciences, The Weizmann Institute of Science, 7610001 Rehovot, Israel. [9]Department of Life Sciences and the National Institute for Biotechnology in the Negev, Ben-Gurion University of the Negev, 8499000 Beer-Sheva, Israel. [10]These authors contributed equally: Jie Li, Haonan Zhang. [11]These authors jointly supervised this work: Yingang Feng, Ping Zhu. ✉e-mail: fengyg@qibebt.ac.cn; zhup@ibp.ac.cn

group I σ factors in the regions from $\sigma_{1.2}$ to $\sigma_4$, and, therefore, present essentially the same structures in the $\sigma_2$, $\sigma_3$, and $\sigma_4$ domains[11,12]. The group III σ factors lack both $\sigma_{1.1}$ and $\sigma_{1.2}$ and show weaker interactions between $\sigma_4$ and the −35 element than those of the group I and II σ factors[13,14]. The group IV σ factors contain only $\sigma_2$- and $\sigma_4$-domains, which bind to RNAP and promoter DNA in a similar strategy to those of the other groups, but the detailed interactions between the group IV σ factor and the promoter DNA are quite different from the interactions of the other groups[15–17]. These interactions are of great importance for the recognition of a consensus sequence of the −35/−10 elements by the group IV σ factors.

The $\sigma^I$ (SigI) factor is a unique $\sigma^{70}$ that is widespread in Bacilli and Clostridia[20–24]. It contains a $\sigma_2$-domain for recognition of the −10 element but lacks the $\sigma_4$-domain that recognizes the −35 element[25]. $\sigma^I$ was initially classified into $\sigma^{70}$-family group III[26] but later considered an ECF-like σ-factor, since its C-terminal domain (SigIC) was suspected of playing a recognition role for the −35 element despite its lack of sequence homology with $\sigma_4$[27,28]. $\sigma^I$ factors are involved in the heat shock response, iron metabolism, virulence, and carbohydrate sensing[21,24]. Multiple paralogues of $\sigma^I$ and cognate anti-$\sigma^I$ factors (RsgIs) have been found, and these $\sigma^I$-anti-$\sigma^I$ operons were shown to regulate component expression of cellulosomes, the multienzyme complexes that mediate efficient cellulose degradation[20,24,29]. These RsgIs contain an exocellular carbohydrate-binding module, positioned to sense the extracellular polysaccharide substrate[30], a periplasmic domain that accommodates an autoproteolytic event for signal transduction[31–33], a transmembrane helix, and a cytoplasmic inhibitory domain that binds to SigI[23]. Promoter sequences recognized by the σIs contain an A-tract motif and a CGWA motif in the −35 and −10 elements, respectively[27,28]. $\sigma^I$ paralogues exhibited distinct promoter-specificity, considered to be related to an upstream region of the A-tract motif[27,28]. Although the N- and C-terminal $\sigma^I$-domains presumably recognize promoter −10 and −35 elements, respectively, it is unknown how they specifically recognize promoter DNA[23,25,27]. The structure of $\sigma^I$ in an active state (in complex with RNAP) is thus needed to elucidate the mechanism of specific promoter recognition by multiple $\sigma^I$s.

Here, we determined high-resolution cryo-EM structures of RNAP-σ-promoter complexes (transcription-ready open complexes, RPo complexes) for two *C. thermocellum* $\sigma^I$s. Structural analysis and functional validation revealed the unique promoter recognition mode and molecular mechanism of specificity for $\sigma^I$ paralogues, which differ from all other known groups of $\sigma^{70}$ factors.

## Results

### Overall structure of RPo-$\sigma^I$

RPo complexes RPo-SigI1 and RPo-SigI6 were reconstituted using purified *C. thermocellum* RNAP core enzyme, purified recombinant *Escherichia coli* SigI1/SigI6, and synthesized P1/P6 DNA scaffolds (Fig. 1A and Fig. S1). The RPo-SigI1 and RPo-SigI6 structures were determined using single-particle cryo-electron microscopy (cryo-EM) (Fig. S2). Final structures were refined to 3.0-Å resolution for RPo-SigI1 and 3.3 Å for RPo-SigI6 (Table S1). Cryo-EM structures of the latter two complexes served to resolve the RNAP core enzyme (α, α, β, β′, and ω subunits), the $\sigma^I$, and promoter DNA with well-defined densities and structural stacking (Fig. 1B; Figs. S3 and S4). According to the previous report[27], the location of the transcriptional start sites (TSSs) of P1 and P6 differs, and the obtained RPo-SigI1 and RPo-SigI6 structures showed opened bubbles at different positions, which are not aligned with the experimental TSSs. For simplicity, we uniformly use base numbers from P1's TSS position for both promoters (Fig. 1A). We observed the densities of DNA of the transcription bubble (−12 to +2) of the non-template strand (NT-strand) and the upstream (−43/−42 to −13 in RPo-SigI1/RPo-SigI6, respectively) and downstream DNA duplex (+3 to +15/+16 in RPo-SigI1/RPo-SigI6, respectively). The obtained RPo-$\sigma^I$

structures adopt a closed conformation, by comparison with known bacterial RNAP holoenzyme and RPo structures (Fig. 1C).

The N-terminal SigI6 domain (SigI6N, residues 13–110) is located in the cleft between the RNAP-β lobe and RNAP-β′ coiled-coil (β′CC) with extensive hydrophobic and hydrogen-bond interactions, while the C-terminal SigI6 domain (SigI6C, residues 134–245) forms hydrophobic interactions with the flap-tip helix (βFTH) of the RNAP β subunit (Fig. 1D and Fig. S5). The relative position of SigIN with RNAP-β′CC is similar to that of other $\sigma^{70}$-family $\sigma_2$-domains and β′CC (Fig. S5A), but the detailed interactions are different, resulting in different helix orientations relative to β′CC (Fig. S5B). These differences are caused by non-conserved interacting residues in the different types of σ factors, although those of RNAP-β′CC are highly conserved (Fig. S5D, E). The interacting hydrophobic residues of SigIC with βFTH are completely different from those of the $\sigma_4$-domains of other $\sigma^{70}$ factors (Fig. S5C, F), because SigIC has no sequence homology with $\sigma_4$ and adopts different structural elements in binding βFTH.

Promoter DNA binds to both $\sigma^I$ and the RNAP core enzyme (Fig. 1D). The upstream region of the promoter forms a duplex and the −35 element interacts with SigIC helices α8-α12. The downstream region forms the transcription bubble through extensive interaction with SigIN. SigIN binds to the −10 element, forming the opening of the bubble, and stabilizing the NT-strand DNA. Finally, the NT and template strands form a duplex and exit RNAP from the channel between the clamp formed by RNAP β and β′ subunits.

Although the overall structures of RPo-SigI1 and RPo-SigI6 are similar, some differences are observed when the structures are aligned by their RNAP core enzymes (Fig. 1E). The SigIC domains show a rotation and shift, and the SigI1C-bound −35 element bends more towards RNAP than the promoter-bound SigI6C. The first α-helix of the N-terminal SigI1 and SigI6 domains also showed different orientations.

The overall architecture of the *C. thermocellum* RPo complexes is similar to other known RPo complexes from various bacteria[16,34–36]. However, structural analysis of the $\sigma^I$-promoter interactions (Fig. S6) indicated that the mechanism of recognition is different from other known $\sigma^{70}$ family members, as shown below.

### Interactions between $\sigma^I$ and promoter −35 element

SigIC binds to the −35 element through both its HTH structure formed by helices α11 and α12 in the DNA major groove and the N-terminal part of helix α9 in the minor groove (Fig. 2A, B and Fig. S7). Although the local resolutions of the SigIC-binding region (about 4.5 Å) are lower than the resolution in the RNAP core regions, and the densities of the SigIC side chains are not always clearly observed, the SigIC model structures predicted by Alphafold[37] fit well into the densities, and some of the large side chain residues, such as Phe and Tyr, can be observed with clear side chain densities (Fig. S4), resulting in the construction of reliable models for the SigIC-promoter binding regions. Minor-groove binding in the −35 element has not been observed in other $\sigma^{70}$-family members[11,13,16,34]. Several residues of helix α9 are involved in the interaction with the minor groove. The side chain of H171/H173 in SigI6/SigI1 is inserted into the minor groove, forming hydrogen bonding and stacking interactions with the ribose rings. Adjacent conserved residues, including R172/R174, S174/S176, and K170/K172, interact with backbone phosphates of the double-stranded DNA (dsDNA). These minor-groove-binding residues are conserved in $\sigma^I$ (Fig. S7C), and the SigIC-binding minor groove is formed by the characteristic, essential A-tract region in $\sigma^I$-dependent promoters[27,28]. To confirm the importance of the SigIC minor-groove-binding residues, we analyzed the activity of SigI6 and its mutants using both an in vivo heterologous Bacillus system[27,28,38] and in vitro transcriptional activity assays[39] (Fig. 2D, E). The in vivo heterologous Bacillus system revealed that mutation of H171 to Tyr, Phe, Asn, Ser, or Ala resulted in complete loss of activity, while mutation to Lys or Arg resulted in significantly decreased but detectable activity, since they are minor-

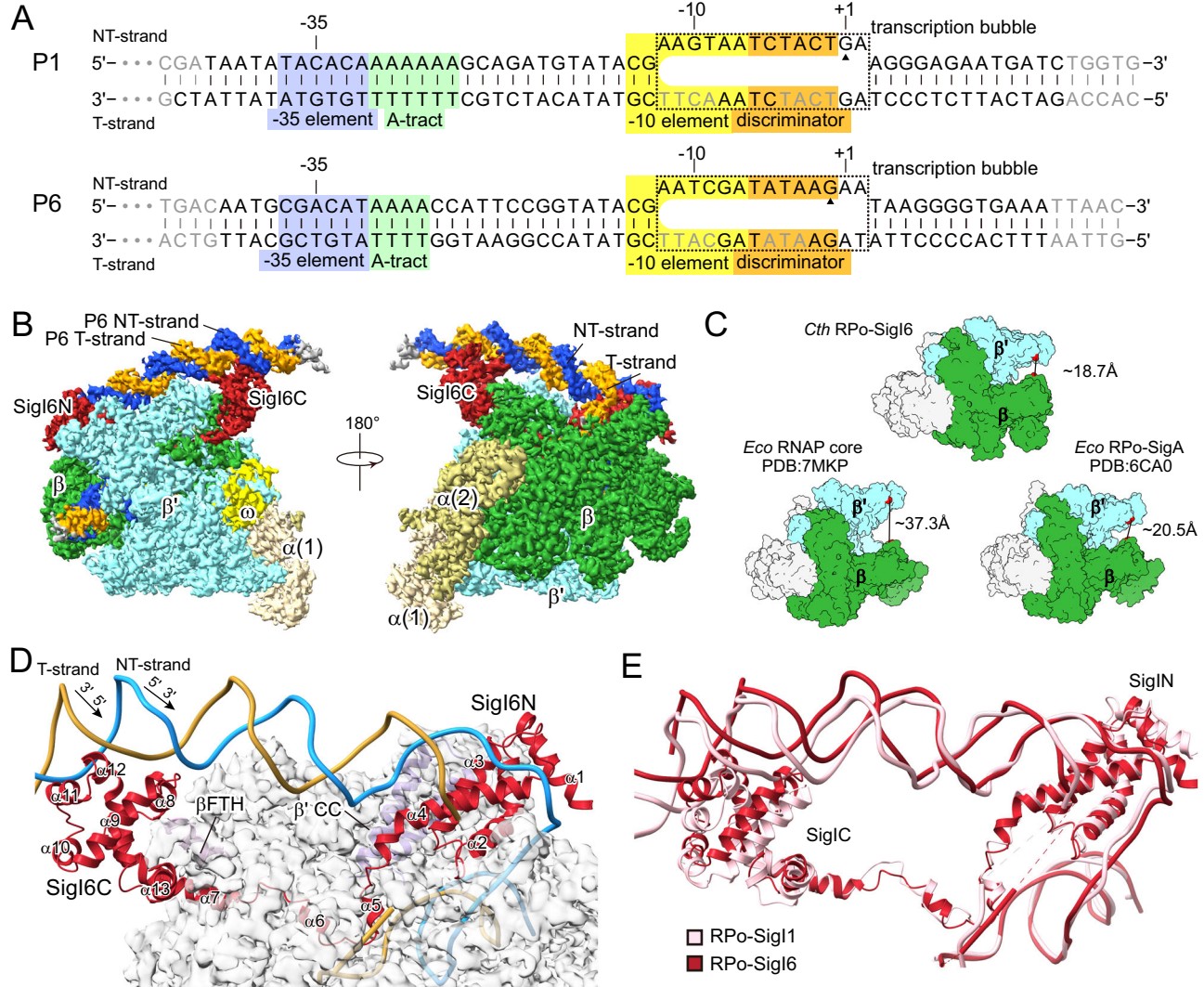

**Fig. 1 | Cryo-EM structures of RPo-SigI1 and RPo-SigI6 from *C. thermocellum*.**
**A** The nucleic-acid scaffolds are used for structure determination. P1 and P6 are SigI1- and SigI6-dependent promoters, respectively. The transcription bubbles observed in the structures are indicated by dashed rectangles. Nucleotides that cannot be modeled in the structures because of poor density are shown in gray fonts. The filled triangles indicate the transcription starting site (TSS) reported in literature[27], which has one nucleotide difference in the alignment. For convenient comparison between two promoters, the nucleotides in P6 are numbered according to the alignment with P1 instead of the TSS of P6. The −35 element, A-tract motif, −10 element, and discriminator are shaded in blue, green, yellow, and orange, respectively. **B** The cryo-EM density map of RPo-SigI6. Each subunit of RPo-

SigI6 and DNA strand is colored differently: β, green; β′, cyan; α1, khaki; α2, dark khaki; ω, yellow; SigI6, red; NT-strand DNA, deep blue; T-strand DNA, orange.
**C** RPo-SigI6 presents a closed conformation of the β-β′ clamp. For comparison, *E. coli* RNAP structures are shown in the open (RNAP core enzyme) and closed (RPo-σ^A) conformations. The clamp distances between residues β G373 and β′ I1290 for *E. coli* RNAP and residues β G242 and β′ I1302 for *C. thermocellum* RNAP are labeled.
**D** The organization of SigI6 (red) and P6 (light blue and orange) on the RNAP core (gray). **E** Comparison of the σ^I and promoter conformations in RPo-SigI1 (pink) and RPo-SigI6 (red). The structures were superimposed by the whole complexes and the subunits of the RNAP cores are not shown.

groove-binding residues observed in A-tract binding proteins[40–42]. Mutation of K170 and R172 also significantly decreased activity, confirming their functional importance. The in vitro transcriptional activity assays also exhibited similar results (Fig. 2E), indicating the functional importance of the minor-groove binding by SigI.

Unlike the minor-groove binding by conserved residues, the major groove of the −35 element was mainly bound by non-conserved SigIC residues. The interacting residues include conserved R215/R217 and non-conserved R219/N221, K200/N202, T203/R205, L204/N206, K221/R223, R214/G216, S213/H215, and E218/G220 of SigI6/SigI1, which form different interactions in the two RPo structures, indicating their selective importance in promoter specificity (Fig. 2B; Figs. S6 and S7A, C). Consistently, the major-groove DNA sequence corresponded to the region of specificity (ROS) proposed previously[27]. Mutation of interacting RPo-SigI6 residues

resulted in the loss of activity (Fig. 2D, E), indicating their roles in promoter binding. Comparison of SigI6C and SigI1C revealed slight differences in helix orientations (Fig. S7B), but showed significant shift and rotation relative to the RNAP core enzyme (Fig. 1E).

The DNA-binding mode of SigIC differs from that of the σ^70-family σ_4-domain, which binds to the major groove only[11,13,16,17,34,43]. The additional minor-groove binding results in significantly larger interface area (952 Å²) between SigIC and promoter versus that between σ_4-domain and promoter (e.g., 769 Å² of σ^H from *M. tuberculosis* and 530 Å² of σ^A from *B. subtilis*). Furthermore, the binding modes of SigIC and σ_4 with the major groove are completely different. Previous studies indicated that SigIC would show steric hindrance if it would adopt a dsDNA binding conformation similar to that of the σ_4-domain of ECF σ-factors[23,44]. The RPo-σ^I structures indeed revealed that although SigIC interacted with the major groove via its HTH structure (α11 and

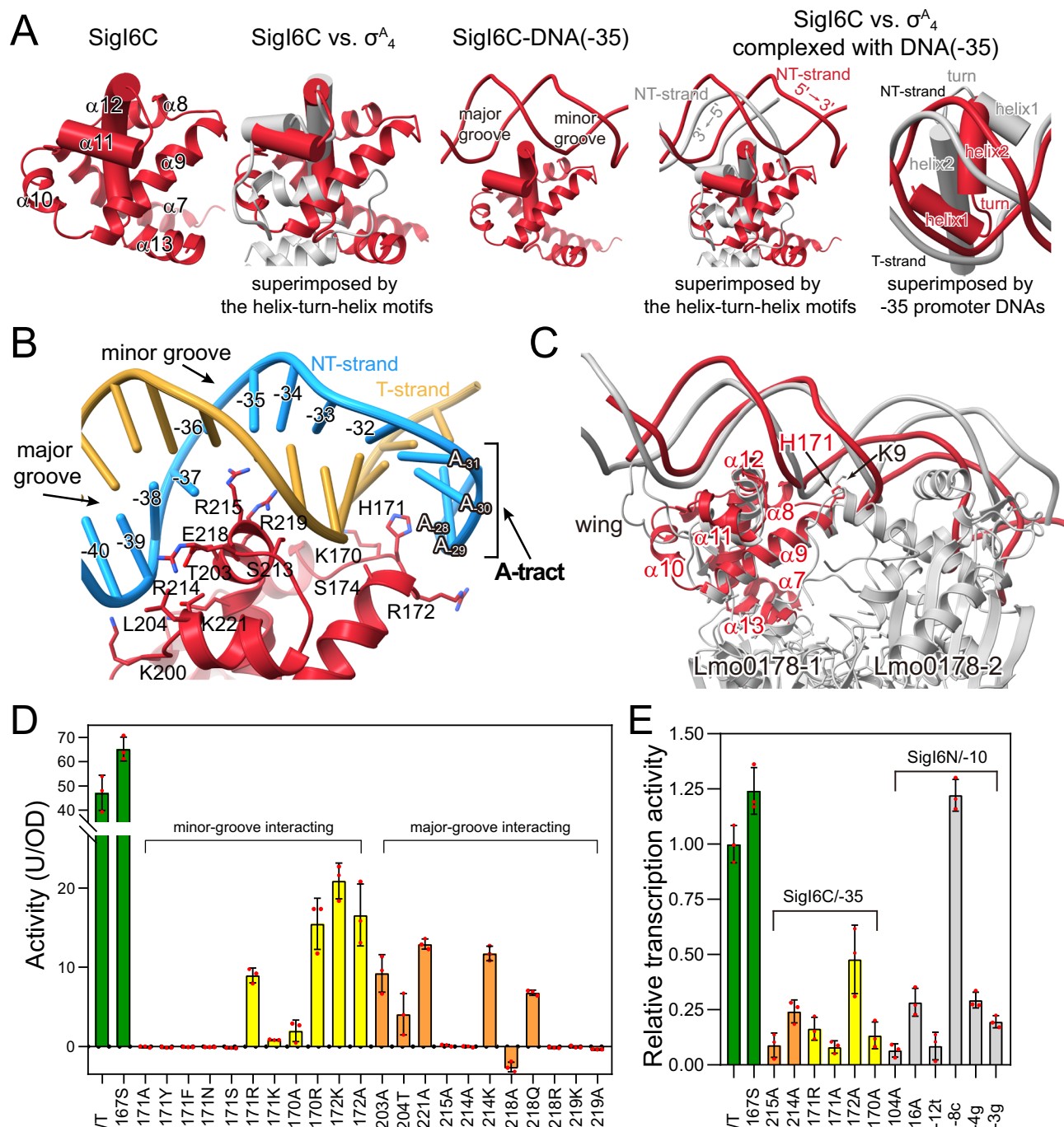

**Fig. 2 | Interactions between the promoter DNA −35 element and SigI6 in the RPo-SigI6 structure. A** Comparison of the DNA-binding modes of the SigI6C domain (red) and the σ₄ domain in σᴬ from *Bacillus subtilis* (PDB 7CKQ, gray). The helices in the helix-turn-helix (HTH) motifs are shown as cylinders. The two panels on the right demonstrate that when the structures are superimposed by the HTH motif, the respective NT strands of the DNA run in opposite directions (indicated by 5′→3′); when the structures are superimposed by the promoter DNAs, the HTH motifs exhibit a ~180° rotation. **B** The detailed interactions in major and minor grooves of the −35 element DNA. Residues involved in the interactions are shown as sticks. SigI6C, red; NT-strand DNA, light blue; T-strand DNA, orange. **C** Comparison of the DNA-binding modes in the SigI6C-promoter (red) recognition and that of a transcription repressor Lmo0178-operator DNA complex (PDB 5F7Q, gray) from

*Listeria monocytogenes*. Lmo0178 is a dimer in the structure and one Lmo0178 molecule (Lmo0178-1) is superimposed with SigI6C. Residues H171 in SigI6 and K9 in Lmo0178, which similarly penetrate the minor groove are shown as sticks with labels. Lmo0178 contains a wing loop, which additionally binds to the upstream minor groove. **D** Activities of SigI6 mutants measured by the *Bacillus subtilis* heterologous reporter system. **E** In vitro transcriptional activities of SigI6 mutants and promoter mutants. The bars are filled with the following colors: green, wild-type and C167S mutant; yellow and orange, mutants for residues potentially interacting with minor and major grooves of the −35 element, respectively; gray, mutants of the −10 element or residues potentially interacting with the −10 element. Data are presented as mean values ± SD, and *n* = 3 biological replicates in **D** and **E**. Source data of **D** and **E** are provided as a Source data file.

α12), its position exhibits a ~180° rotation compared with that of the σ$_4$-domain (Fig. 2A). This rotation not only resolves the potential steric clash but also allows the N-terminal part of helix α9 to fit into the minor groove forming an additional DNA-binding interface, representing a unique binding mode among the known σ$^{70}$ factors. A Dali search[45] revealed that the DNA-binding mode of SigIC is similar to the winged HTH domain of transcriptional factors, among which an ROK-family repressor Lmo0178[46] shows high similarity (Fig. 2C). HTH motifs of SigIC and Lmo0178 similarly bind to the major groove and a positively charged residue (His171/Lys9 in SigI6/Lmo0178, respectively) on an N-terminal helix that penetrates the downstream minor groove (Fig. 2C). However, as opposed to SigIC, Lmo0178 is dimeric and binds to a palindrome sequence, and a β-loop wing binds to the upstream minor groove.

In summary, SigIC has a unique −35 element recognition mode formed by two features: the conserved minor-groove A-tract binding lacking in σ$_4$-promoter recognition, and non-conserved major-groove ROS-binding by the HTH motif, which presents a ~180° rotation compared to the σ$_4$-HTH motif. Therefore, σ$^I$-promoter recognition of the −35 element differs completely from that of the σ$_4$-domain of other σ$^{70}$ factors.

### Interactions between σ$^I$ and the −10 element

The SigIN domain adopts an oval structure formed by three helices α2-α4, similar to the σ$_2$-domain of other σ$^{70}$ factors, and helix α1 is attached to one head of the oval, somewhat similar to the second helix of the σ$_{1.2}$ region (σR1.2) of groups I and II (Fig. 3A). Similar to other σ$_2$-domains, SigIN opens the duplex of the −10 element to form the transcriptional bubble, mainly through helix α4. SigIN also binds the NT-strand through α1, α2, and Loop3 (connecting α3 and α4), thus stabilizing the unwound transcription bubble. The bubble size (number of unpaired nucleotides) is 14 bp, similar to that (13–15 bp) opened by groups I-III σ$^{70}$ factors[9,11,47–49] but different from that (12 bp) of ECF σ-factors[15–17]. Although the overall structure is similar to the σ$_2$-domain of other σ$^{70}$ factors, the detailed comparison showed unique interactions between SigIN and promoter DNA for specific promoter recognition (Fig. 3A), as described below.

The conserved C$_{-14}$G$_{-13}$W$_{-12}$A$_{-11}$ motif (CGAA in both P1 and P6) at the −10 element is recognized by helix-α4 residues. Paired C:G(−14) and G:C(−13) interact with R104/R108, D101/D105, R97/R101, and R98/R102 of SigI6N/SigI1N, and A$_{-12}$ initiates bubble formation. R97/R101 in SigI6/SigI1 serves as a wedge to disrupt stacking between positions −13 and −12. A$_{-11}$ is inserted into the protein pocket, formed by N-terminal A78/K82, K83/K87, D80/D84, and H84/G88 of Loop3 and N-terminal conserved F90/F94 of helix α3 (Fig. 3B and Fig. S8A). Mutation in conserved residues that bind C:G(−14), G:C(−13), and A$_{-11}$ resulted in near-complete activity loss, except for the D80A mutation (Fig. 3C). Non-conserved H84 does not play a key role in specific recognition. A$_{-12}$ showed extensive interaction with identical residues R97/R101, E74/E78, F90/F94, and Q93/Q97 in SigI6/SigI1, but R97/R101, F90/F94 and Q93/Q97 are only partially conserved in the other σ$^I$s (Fig. S8B). Therefore, we suspect that non-conserved W$_{-12}$ may partially contribute to specific promoter recognition for different σ$^I$s. A previous study showed that SigI3 binds the CGTA motif, and CGTA-to-CGAA mutation of the SigI3-dependent promoter P$rgl11A$ resulted in a complete loss of SigI3 recognition[28]. The CGTA mutant (A$_{-12t}$) of P$sigI6$ cannot be recognized by SigI6 in the heterologous Bacillus system (Fig. 3C), and the in vitro transcription activity of the CGTA mutant also decreased significantly (Fig. 2E), indicating the importance of W$_{-12}$ in σ$^I$ specificity.

All unpaired bases of the NT-strand DNA in the bubble (from −12 to +2) in the two RPo-σ$^I$ structures turn outward with abundant π-stackings between successive bases, and bases from −12 to −3 form extensive interactions with SigIN (Fig. 3A, B and Fig. S8A). This is a unique structural feature in known RNAP-σ complexes, since only part

of the NT-strand bases in the bubble flip out in other group σ$^{70}$-RNAP complexes (Fig. 3A)[11,16,43]. According to sequence alignment (Fig. S8B), residues binding to −10 element downstream bases are largely non-conserved in σ$^I$. The −10 to −7 downstream promoter region together with A$_{-11}$ showed extensive interactions with Loop3—the "specificity loop" in ECF-σ[16,50]. The latter loop specifically recognizes the −11 base in the X$_{-14}$G$_{-13}$T$_{-12}$Y$_{-11}$ (X = C,G; Y = A,T,C) motif[3,25], which spatially corresponds to T$_{-10}$ of P$sigI6$. Since this position is not conserved in σ$^I$-dependent promoters, we investigated whether Loop 3 plays a specificity role in the different σ$^I$s. Mutation of T$_{-10}$ of P$sigI6$ into different nucleotides resulted in different effects: T$_{-10}$c showed much higher activity than wild-type P$sigI6$, while T$_{-10}$g and T$_{-10}$a showed complete and partial loss of activity, respectively. Similarly, mutations H84 and S85 of SigI6, according to the mutation pattern in SigI1 (H84G/S85Y), SigI2 (H84N/S85M) and SigI3 (H84N/S85G), resulted in diverse effects (Fig. 3C). The latter inconsistent results indicated that the downstream region of the CGWA motif is likely a modulator of promoter activity but does not serve as a specificity determinant for the different σ$^I$s.

### Structural comparison of active σ$^I$ in the RPo complex and RsgI-bound inhibited σ$^I$

Our previous study showed that RsgI specifically binds to the C-terminal domain of cognate σ$^I$ to inhibit σ$^I$ activity and that the interface contains both conserved and non-conserved residues[23]. Nevertheless, how this interaction inhibits σ$^I$ activity is unclear. The structure of the RPo complex revealed only slight conformational changes between the active and inhibited states of SigIC (Fig. 4A), and the same surface binds to RNAP and RsgI (Fig. 4B, C), thus indicating that RsgI inhibits σ$^I$ activity by competitive binding. SigIC binds βFTH through conserved hydrophobic surface residues (Fig. 4B and Fig. S7C), which partly overlap with the RsgI-binding residues (Fig. 4C and Fig. S7C). However, the interface area of the RsgI1-SigI1 interaction (1056 Å$^2$) is much larger than that of SigI1C-βFTH (800 Å$^2$), which might explain why the conserved σ$^I$-RNAP interaction is inhibited by the non-conserved interaction with RsgI.

## Discussion

Despite more than 20 years of study of the σ$^I$s since their discovery[22], their classification remains confusing. σ$^I$s were initially classified as σ$^{70}$-family group III[26] and later reclassified as "ECF-like"[27,28] rather than ECF σ factors[3,25,51]. Our structures of the RPo-σ$^I$ complexes indicated that σ$^I$ is indeed a unique type of σ factor that cannot be classified into canonical groups of the σ$^{70}$ family. Several features distinguish σ$^I$ from the other groups. In this context, σ$^I$ has a σ$_2$-domain that contains part of σR1.2 which only exists in members of groups I and II. In addition, the RPo-σ$^I$ complex contains a bubble size similar to those of groups I-III. However, σ$^I$ lacks a σ$_3$ domain which exists in the latter groups (Fig. 5A). Moreover, the −10 element binds to SigIN with more flipped-out bases than those of other σ$^{70}$-promoter complexes (Fig. 5B). Finally, although SigIC is responsible for recognition of the −35 element and is functionally similar to the σ$_4$-domain of the other σ$^{70}$ factors, it is completely different, both in terms of structure and DNA-binding mode (Fig. 5B). Therefore, σ$^I$ factors represent a distinct member of the σ$^{70}$-family σ factors, thus highlighting the diversity of bacterial transcription.

Intriguingly, the C-terminal σ$^I$ domain binds to −35 DNA with a large binding surface that penetrates both major and minor grooves of the promoter DNA. The minor- and major-groove regions correspond to the previously identified A-tract and region of specificity[27], respectively, and the present study provides a structural basis for the function of the two regions. The manner of minor-groove binding by a single positively charged residue has been widely observed in DNA-binding proteins for specific A-tract or AT-rich DNA recognition[52,53]. Major-groove-binding by SigIC is similar to winged helix-turn-helix (HTH) domains of transcription factors[46,54]. However, evolutionary

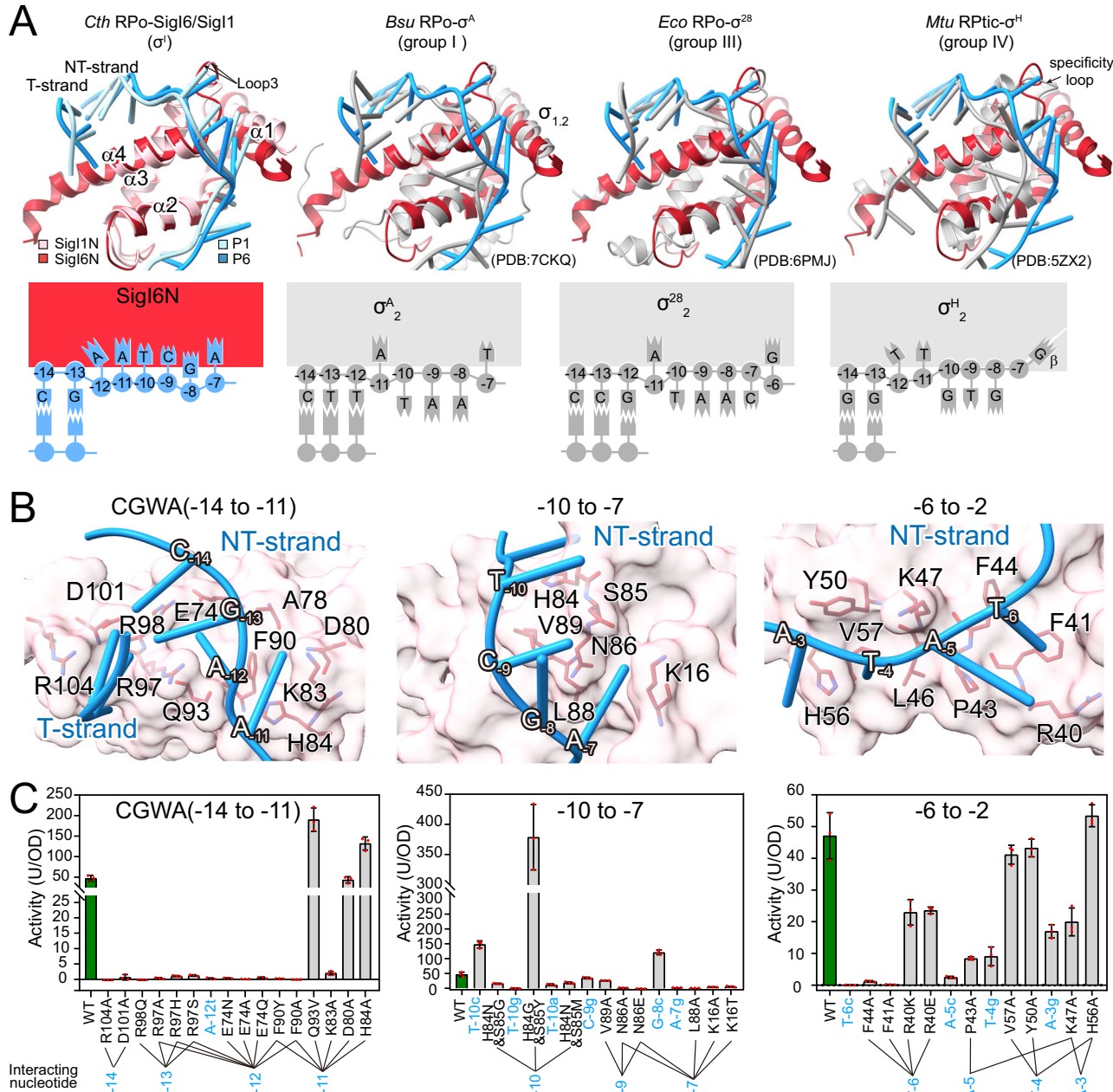

**Fig. 3 | Interactions between promoter DNA −10 element and SigIN in the RPo-σ$^I$ structures. A** Comparison of the DNA binding by the σ$_2$ domain of SigI, group I, group III, and group IV σ factors. SigI1N and SigI6N are shown in pink and red, respectively, and other group σ factors are shown in gray. The schematic diagrams of the promoter −10 element recognition by different types of σ factors are shown at the bottom, indicating more interactions between SigIN and the −10 element (blue) than those between other group σ factors and the −10 elements (dark gray). **B** Interactions between *C. thermocellum* SigI6 (pink) and transcription-bubble DNA

(blue). SigI6 is rendered as surfaces, and the residues involved in protein−DNA interactions are shown as red sticks. **C** Activities of various SigI6 mutants in the SigI6N region or P*sigI6* mutants in the −10 region, measured by the *B. subtilis* heterologous reporter system. Green, wild-type and C167S mutant; gray, mutants of the −10 element or residues potentially interacting with the −10 element. Data are presented as mean values ± SD, and *n* = 3 biological replicates in **C**. Source data of **C** are provided as a Source data file.

relationships were lacking between σ$^I$ and the transcriptional factors upon comparing their homologous sequences in various bacteria. The similarity is likely caused by the convergent evolution of two different proteins for DNA binding.

The two structures reported here provide insight into the specificity of different σ$^I$ paralogues in one bacterium. Non-conserved residues in the HTH motif of the SigIC domain specifically bind to the ROS in the promoter −35 element. In addition, the −12 nucleotide in the promoter −10 element plays a role in the specificity. Its downstream nucleotides show extensive interactions with σ$^I$ and probably

modulate the activities for each specific gene. The numbers of interacting residues in σ$^I$ and interacting nucleotides in the promoter are much higher than those of the ECF σ factors, which may explain why one bacterium can maintain so many (up to sixteen) σ$^I$s for regulation with specificity[27]. Since the σ$^I$s in *C. thermocellum* are responsible for regulating the expression of cellulosome components−thus comprising a potential "treasure-trove for biotechnology"[55,56], the promoter recognition mechanism revealed in this study provides the basis for future engineering of cellulosome production in cellulosome-producing bacteria[57]. Furthermore, the unique binding mode and

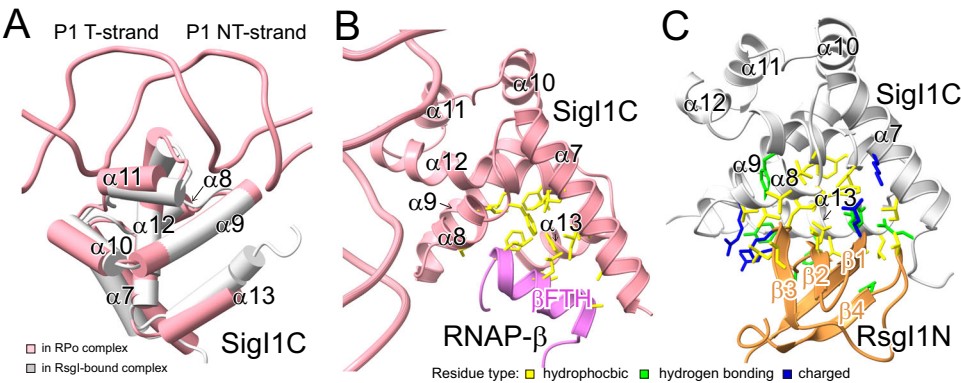

**Fig. 4 | Comparison of SigI1C in the active (i.e., in RPo complex) and the inactive (i.e., RsgI-bound) states. A** Comparison of SigI1C structures in the active (pink) and inactive (PDB 6IVU, gray) states. **B** Interaction between SigI1C (pink) and RNAP βFTH (light purple). **C** Interaction between inactive SigI1C (gray) and RsgI1N (orange). In **B** and **C**, residues involved in hydrophobic, hydrogen bonding, and electrostatic interaction are shown as yellow, green, and blue sticks, respectively.

specificity mechanism of the σ$^I$s provide new possibilities to design regulators in synthetic biology for the design of orthogonal genetic switches and regulators[8,58].

## Methods

### Purification of RNAP core enzyme from *C. thermocellum*

The strains used in this study are listed in Table S2. The plasmids and primers used in this study are listed in Supplementary Data 1 and Supplementary Data 2, respectively. The *C. thermocellum* strain for the purification of RNAP was constructed using the previously developed homologous recombination method[59]. Specifically, a strong constitutive promoter P$_{2638}$[60] and an N-terminal His×10-tag were inserted before the RNAP β' gene (*clo1313_0314*) [https://www.ncbi.nlm.nih.gov/gene/12420012] in *C. thermocellum* strain Δ*pyrF*[59]. The homology arms Bp-UP and Bp-DN and the promoter P$_{2638}$ were amplified by PCR from *C. thermocellum* DSM1313 genomic DNA. The DNA fragments were ligated by either overlapping PCR or restriction enzyme digestion and T4 ligation, and finally the homologous recombination plasmid pHKm2-homo-5'Betap was obtained (Fig. S1A). The plasmid was transformed into *C. thermocellum* strain Δ*pyrF* by electroporation[59], generating the mutant DSM1313::P$_{2638}$-His$_{10}$-β' after two rounds of screening. Transformants containing the plasmid pHKm2-homo-5'Betap were first screened on semi-solid GS-2 medium containing 3 µg/mL thiamphenicol (Tm). Then, the obtained transformant was screened with 10 µg/mL 5-fluoro-2-deoxyuridine (FUDR)-supplemented uracil auxotrophic MJ medium to generate the target mutant after homologous recombination. The mutant was verified by colony PCR and sequencing. *C. thermocellum* strains were routinely cultured anaerobically at 55 °C in GS-2 medium, supplemented with 5.0 g/L cellobiose as carbon source.

The RNAP core enzyme was directly purified from DSM1313::P$_{2638}$-His$_{10}$-β'. The cells were grown anaerobically at 55 °C in 50 L GS-2 medium supplemented with 5 g/L glucose as a carbon source. When the optical density at 600 nm (OD$_{600nm}$) reached 1.2 - 1.8, cells were collected by centrifugation at 10,200 *g* for 30 min. The cell pellet was suspended in 1.5 L buffer A (20 mM Tris-HCl pH 8.0, 500 mM NaCl, 30 mM imidazole, 5% (v/v) glycerol, and protease inhibitor cocktail (Roche)) and lysed by ultrasonication. The lysate was centrifuged at 15,000 *g* for 50 min at 4 °C, the supernatant was then loaded onto a 40-mL His-Trap FF affinity column (GE Healthcare Life Sciences) pre-equilibrated with buffer A, and RNAP was eluted by buffer B (20 mM Tris-HCl pH 8.0, 500 mM NaCl, 500 mM imidazole, 5% (v/v) glycerol). The complex was further purified using a 5-mL Hi-Trap Heparin column (GE Healthcare Life Sciences) pre-equilibrated with buffer C (20 mM Tris-HCl pH 8.0, 100 mM NaCl, 2 mM DTT, 0.2 mM EDTA, 5% (v/v) glycerol, and protease inhibitor cocktail (Roche)), and the RNAP was eluted with buffer D (20 mM Tris-HCl pH 8.0, 1 M NaCl, 2 mM DTT,

0.2 mM EDTA, 5% (v/v) glycerol) by a linear gradient. The fractions containing RNAP were collected and loaded on a Source Q column (GE Healthcare Life Sciences). After elution by a linear gradient of NaCl to the final concentration of 1 M, the fractions containing RNAP core enzyme were collected, concentrated to 3 mg/mL, and stored at −80 °C. The subunits of the RNAP core enzyme in the purified proteins were identified by SDS-PAGE.

### Purification of recombinant SigI1 and SigI6

The expression and purification of SigI1, SigI6, and mutants of SigI6 (C167S, R215A, R214A, H171R, H171A, R172A, K170A, R104A, and K16A) in *Escherichia coli* followed the procedures for SigI1 reported in a previous study[23]. Briefly, the gene fragments encoding full-length SigI1 and SigI6 were cloned into the pET28a-SMT3 plasmid, generating the plasmids pET28a-SMT3-SigI1 and pET28a-SMT3-SigI6. Each mutant of SigI6 was constructed by site-directed mutagenesis using the QuikChange method. All the plasmids were transformed into *E. coli* BL21 (DE3) for protein expression. The wild-type SigI6 showed poor stability during the purification, and the mutant SigI6-C167S showed much better stability. Therefore, SigI6-C167S was purified and used in the structural study. The recombinant proteins were first purified by a nickel-affinity column and then purified further by size-exclusion chromatography with a HiLoad 16/600 Superdex 75 column with buffer E (20 mM Tris-HCl pH 8.0, 150 mM NaCl, 2 mM DTT, 0.2 mM EDTA, 5% (v/v) glycerol, 10 mM MgCl$_2$). The purity of recombinant proteins was detected using SDS-PAGE (Fig. S10).

### Nucleic acid scaffolds

Double-stranded nucleic acid scaffolds for the cryo-EM study of RPo-SigI1 and RPo-SigI6 were prepared from synthetic oligos (Table S3) by annealing the DNA (heating at 95 °C for 5 min and then allowing the DNA to cool slowly to room temperature). The annealing buffer contains 20 mM Tris-HCl pH 8.0, 200 mM NaCl, and 10 mM MgCl$_2$.

Double-stranded nucleic acid scaffolds for the fluorescence-detected in vitro transcription assay were prepared by PCR using pUC19-PsigI6-Mango-tR2 as a template. DNA sequences and primers used for the in vitro transcription assay are listed in Tables S4 and Supplementary Data 2, respectively.

### Reconstitution of the RPo-σ$^I$ complex

To reconstitute the RPo-SigI1 and RPo-SigI6 complexes for cryo-EM, the purified *C. thermocellum* RNAP core enzyme, purified recombinant SMT3-SigI1 or SMT3-SigI6-C167S, and annealed nucleic-acid scaffold were mixed at 1:3:1.3 molar ratio and incubated at 4 °C overnight. The reconstituted RPo-σ$^I$ complex was further treated with ULP1 protease

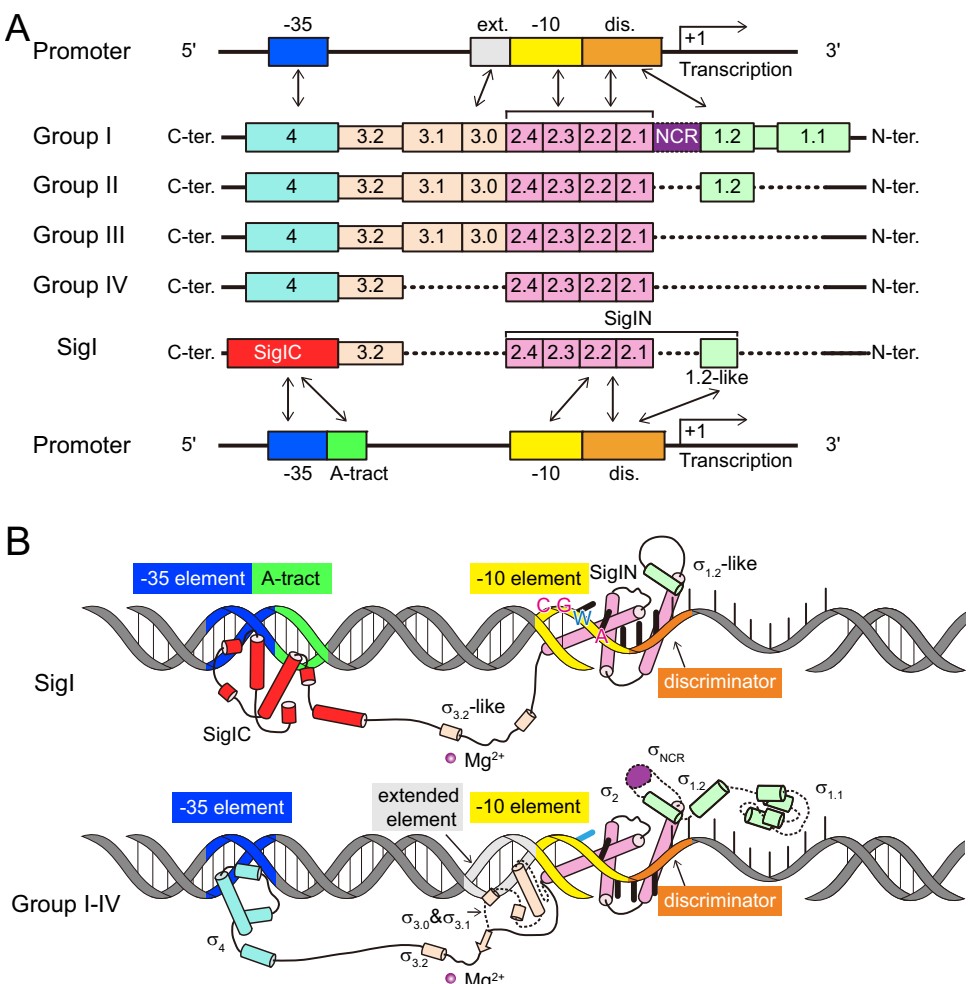

**Fig. 5 | Schematic diagrams of the promoter recognition by σ$^I$ and by the four groups of σ$^{70}$ family σ factors. A** Domain organization of the different groups of the σ$^{70}$ family σ factors. The promoter regions recognized by the different domains are indicated by arrows. **B** Cartoon models of the promoter binding mode by σ$^I$ (top) and other σ$^{70}$ factors (bottom). For the cartoon of groups I-IV, the domains existing in only part of the groups are shown as dashed lines. The −35 element, A-tract, −10 element, and discriminator DNAs are colored in blue, green, yellow, and orange, respectively. The SigIC, σ$_4$, σ$_3$, σ$_2$, σ$_1$, and NCR domains of σ factors are colored in red, cyan, khaki, pink, light green, and purple, respectively.

to remove the SMT3 tag of SMT3-SigI1/6. The RPo-σ$^I$ complexes were concentrated to 500 μL and then purified using a Superdex 200 Increase 10/300 GL column in buffer E. The fractions of the RPo-σ$^I$ complex were collected and concentrated for cryo-EM sample preparation. The subunits and DNA scaffolds of RPo complexes were identified by SDS-PAGE and Native-PAGE (Fig. S1E, F).

**Cryo-EM grid preparation**
The purified samples (12–24 mg/mL protein) were mixed with 8 mM CHAPSO (final concentration) and 0.1 mM DTT. Quantifoil R1.2/1.3 holey carbon grids were glow-discharged for 90 s before the application of 3 μL of the sample. After blotting for 6–8 s with a blot force of 2 N, the grids were plunge-frozen in liquid ethane using an FEI Vitrobot Mark IV (FEI, Hillsboro) with 95% chamber humidity at 10 °C.

**Cryo-EM data acquisition and processing**
The grids were imaged using a 300-keV Titan Krios equipped with a K2 Summit direct electron detector (Gatan) and a GIF quantum energy filter (slit width 20 eV). Data were collected at a nominal magnification of ×22,500 (1.04 Å/pixel) with a dose rate of 8 electrons/pixel/s on the sample (-7.8 electrons/pixel/s on the detector). All images were recorded using Serial EM[61] with super-resolution counting mode for

7.6 s exposures in 32 subframes to give a total dose of 60 electrons/Å$^2$ with defocus range of −1.5 to −2.5 μm.

Motion correction and CTF estimation of cryo-EM movies were performed using Warp[62], and particles were picked using an instance of Warp's neural network retrained on 100 selected micrographs RPo-SigI1 data sets and RPo-SigI6 data sets. Particles were extracted in Warp and subsequently classified in cryoSPARC[63].

For the RPo-SigI6 dataset, the initial model was generated by *Mycobacterium tuberculosis* wild-type RNAP holoenzyme/RbpA/CarD/Sor/AP3−RP2 class (EMD-22575) as a template to 3D classify the particles using cryoSPARC heterogeneous refinement. The best class was selected as the reference to classify the particles for 3D classification with alignment in RELION[64]. A collection of 120591 particles was selected to perform autorefinement. Focused classification (without alignment) of the SigI6C terminal was performed to improve the local density of the SigI6 and binding DNA. To further clean the dataset, CryoDRGN[65] was used to classify particles, and three similar classes were selected to perform the non-uniform refinement. The map was estimated to be at a resolution of 3.58 Å in RELION, and further processing by density modification with the ResolveCryoEM program[66] improved the map quality and resolution to 3.36 Å.

For the RPo-SigI1 dataset, the extracted particles were first 3D- and 2D-classified in cryoSPARC to discard poor particles. The remaining

particles were then subjected to 3D classification in Relion and refinement in cryoSPARC to obtain a reconstructed map. To improve the local density of SigI1 and binding DNA, focused classification (without alignment) of the SigI1C terminus was performed in Relion. All particles in the best class during the focused classification were then subjected to non-uniform refinement in cryoSPARC, resulting in a map with an overall resolution of 3.03 Å. Post-processing of the density map generated during refinement was performed using DeepEMhancer[67]. Local resolution estimations were calculated within RELION. The procedures for Cryo-EM structure determination of RPo-SigI1 and RPo-SigI6 are shown in Fig. S2.

## Model building and refinement

The final cryoEM map for RPo-SigI1 and RPo-SigI6 complexes was used for initial model building. The crystal structure of *Mycobacterium tuberculosis* RPtic-σ^H complex structure (PDB ID 5ZX2) [https://www.rcsb.org/structure/5ZX2] was placed in the cryoEM maps of the RPo-SigI1 and RPo-SigI6 complexes, by rigid-body fitting with UCSF Chimera[68]. The RNAP subunits in RPo-σ^I complexes were manually rebuilt into the cryoEM map referring to the fit RPtic-σ^H structure. The individual models of SigI1 and SigI6 were built referring to the structure predicted by Alphafold[37]. The model was completed and manually adjusted residue-by-residue with real-space refinement in Coot[69], and then followed by real-space refinement in PHENIX[70]. The models were visualized with UCSF Chimera, UCSF ChimeraX[71], and PyMOL (http://www.pymol.org/).

## *Bacillus subtilis* strain construction

A heterologous *B. subtilis* host system was constructed to study the σ^I-dependent promoter activities, referring to the published system which has been successfully used to study the activities of σ^Is from *C. thermocellum* and *Pseudobacteroides cellulosolvens*[27,28]. Plasmids and primers in the present work are listed in Supplementary Data 1 and Supplementary Data 2, respectively. Two plasmids pULacZ and pAX05 were constructed to integrate the *lacZ* reporter gene and the *C. thermocellum sigI6* gene into the *amyE* and *sigI-rsgI* loci, respectively, of *B. subtilis*. The plasmid pAX05 (Fig. S9A) was constructed from plasmid pAX01 carrying an erythromycin (Erm) resistance cassette and the xylose-inducible promoter P*xylA*[27,71]. The upstream (1011 bp) and downstream (1011 bp) regions of *B. subtilis sigI-rsgI* operon were used as the homologous recombination arms and amplified using primer pairs sigI-F1/sigI-R1 and rsgI-F1/rsgI-R1, respectively, from the genomic DNA of *B. subtilis* strain 168. The *C. thermocellum sigI6* gene was amplified using the primer pair Bs-sigI6-F1/ Bs-sigI6-R1. Then the DNA fragments of the homologous recombination arms, the promoter P*xylA*, the *sigI6* gene, and the linearized pAX01 vector generated by PCR were ligated simultaneously with the One Step Cloning Kit (Vazyme), thereby obtaining the pAX05 plasmid. The plasmid pULacZ was constructed from the pUC19 vector (Fig. S9B). A spectinomycin (Spc)-resistance gene as a selectable marker[72] was amplified from plasmid pLH-16 (provided by Mr. Hui Li, Qingdao Institute of Bioenergy and Bioprocess Technology). The upstream (1074 bp) and downstream (825 bp) regions of *B. subtilis amyE* were used as the homologous recombination arms and amplified using primer pairs amyE-F1/amyE-R1 and amyE-F2/amyE-R2. The reporter *lacZ* gene was amplified from the *E. coli* genome using primer pairs lacZ-F1/lacZ-R1. The promoter P*sigI6* was amplified from *C. thermocellum* genomic DNA. Then the DNA fragments and the linearized pUC19 vector generated by PCR were ligated, generating the pULacZ plasmid. The plasmids containing the mutation of SigI or P*sigI6* were obtained by site-directed mutagenesis using the primer pairs listed in Supplementary Data 2. All the used strains are listed in Table S2.

 *B. subtilis* strains were grown on LB, SM1, or SM2 media[73] at 37 °C. The competent cells of *B. subtilis* 168 were prepared following the

reported protocol[73]. *B. subtilis* 168 was transformed with pAX05 and pULacZ plasmids successively. The transformants were selected with 3 μg/mL Erm and 100 μg/mL Spc. Chromosomal integration of plasmids by a double-crossover event was confirmed by colony PCR using the primers listed in Supplementary Data 2.

## Promoter activity analysis by the *B. subtilis* reporter system

To measure the β-galactosidase activity of LacZ in the *B. subtilis* reporter system, strain samples were inoculated into MCSE media with Erm and Spc, and the culture was shaken at 250 rpm until $OD_{600nm} = 0.4-0.5$. Then xylose was added to the final concentration of 1% to induce the expression of SigI for 2 h[28,74]. The β-galactosidase activity was analyzed using ortho-nitrophenyl-β-galactoside (ONPG) as the substrate according to the previously described procedures[28]. Briefly, 4 mL of the cell cultures was centrifuged at 5000 $g$ for 10 min, and the cell pellet was washed twice with Z-buffer (60 mM $Na_2HPO_4$, 40 mM $NaH_2PO_4$, 10 mM KCl, 1 mM $MgSO_4$, pH 7.0) and resuspend in 700 μL working buffer (60 mM $Na_2HPO_4$, 40 mM $NaH_2PO_4$, 10 mM KCl, 1 mM $MgSO_4$, and 2.7 mM β-mercaptoethanol, pH 7.0). The cells were lysed by ultrasonication and the lysate was centrifuged at 13,000 $g$ for 10 min. The 100 μL enzymatic reaction system contained different volumes of cell lysate, 10 μL ONPG stocking solution (13.1 mg/mL in double distilled water), and the working buffer to make up a volume of 100 μL. The reaction system was incubated at 37 °C for a certain period and then 40 μL of the reaction solution was added to 200 μL 1 M $Na_2CO_3$ to terminate the reaction. The released 2-nitrophenol (ONP) was measured by determining the absorbance at 420 nm ($A_{420nm}$). One unit of enzyme activity was defined as the amount of β-galactosidase that releases 1 nmol of ONP per minute. The enzymatic activity was normalized with cell density ($OD_{600nm}$).

## Fluorescence-detected in vitro transcription assay

The measurement of transcription activity was conducted by utilizing the significantly enhanced fluorescence of TO1–3PEG-Biotin when the Mango riboswitch is engaged[75], which has been successfully used to study transcriptional activities of various RNAPs[39,76]. Briefly, to measure the transcriptional activity of SigI6 mutants or P*sigI6* mutants, reaction mixtures (20 μL), containing the *C. thermocellum* RNAP core enzyme (final concentration 50 nM), promoter DNA or its mutants (final concentration 50 nM), and SigI6 or its mutants (100 nM) in reaction buffer (50 mM Tris-HCl pH 7.9, 100 mM KCl, 10 mM $MgCl_2$, 1 mM DTT, 5% glycerol, and 0.01% Tween-20), were incubated at room temperature for 10 min. The reactions were initiated by the addition of 2 μL NTP mixture (UTP, ATP, GTP, and CTP; final concentration 0.1 mM of each) and 2 μL TO1-3PEG-Biotin (final concentration 0.5 μM), and the reaction mixture was incubated at 55 °C for 30 min. The fluorescence signals were measured using a plate reader (SpectraMax M2, Molecular Devices) at an excitation wavelength of 510 nm and an emission wavelength of 550 nm.

## Reporting summary

Further information on research design is available in the Nature Portfolio Reporting Summary linked to this article.

# Data availability

The models and cryo-EM maps have been deposited into the Protein Data Bank and the EMDB under accession numbers 8I23 and EMD-35130 for RPo-SigI1 and 8I24 and EMD-35131 for RPo-SigI6, respectively. Other structure data used in this study for analysis (7MKP, 6CA0, 7CKQ, 6MPJ, 5ZX2, 6IVU) are available in the Protein Data Bank. Protein sequences used in this study are available from Uniprot under accession codes A3DBH0 (SigI1) [https://www.uniprot.org/uniprotkb/A3DBH0/entry], A3DH98 (SigI6) [https://www.uniprot.org/uniprotkb/A3DH98/entry]. Source data are provided in this paper.

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

## Acknowledgements

We thank Prof. Yu Zhang and Miss Linlin You (Shanghai Institute of Plant Physiology and Ecology, Chinese Academy of Sciences) for their helpful suggestions during this study. We thank Mr. Hui Li (Qingdao Institute of Bioenergy and Bioprocess Technology, Chinese Academy of Sciences) for providing the plasmid carrying the spectinomycin-resistance gene. We thank Professor Dawei Zhang and Dr. Gang Fu (Tianjin Institute of Industrial Biotechnology, Chinese Academy of Sciences) for their helpful comments during the construction of the *Bacillus subtilis* reporter system. This work was supported by the National Natural Science Foundation of China [32070125 to Y.F., 3217005 to Q.C., 32070028 to Y.-J.L., 32241029 and 31730023 to P.Z.]; the National Key Research and Development Program of China [2021YFA1300100 to P.Z.]; Shandong Energy Institute [SEI S202106 to Q.C., SEI I202106 to Y.F.]; Qingdao Independent Innovation Major Project [21-1-2-23-hz to Y.-J.L.]; Strategic Priority Research Program of the Chinese Academy of Sciences [XDA21060201 to Q.C., XDB37010100 to P.Z.]. E.A.B. is the incumbent of The Maynard I. and Elaine Wishner Chair of Bio-organic Chemistry. The funders had no role in the study design, data collection, and interpretation, or the decision to submit the work for publication.

## Author contributions

Conceptualization: Y.F.; Methodology: J.L., H.Z., D.L.; Investigation: J.L., H.Z., D.L.; Visualization: J.L., H.Z., Y.F.; Funding acquisition: Y.J.L., Q.C., Y.F., P.Z.; Supervision: Y.F., P.Z.; Writing—original draft: J.L., H.Z., Y.F.; Writing—review & editing: Y.J.L., E.A.B., Q.C., Y.F., P.Z.

## Competing interests

The authors declare no competing interests.
