## [Peer Review File · Nature Communications]

Structure of the transcription open complex of distinct σ^l factorsREVIEWER COMMENTS

Reviewer #1 (Remarks to the Author):

The authors report cryo-EM structures of bacterial transcription initiation complexes dependent on the transcription initiation factor sigma-I. The structures show that sigma-I uses a novel DNA binding domain, different from those in other sigma factors, to recognize the promoter -35 region and reveal the architecture and interactions of the novel DNA binding domain.

The results indicate that sigma-I, which previously had been classified as a Group IV sigma factor, should be re-classified as the first member of a new class of sigma factors having the overall organization of Group IV sigma factors, but having a different structural module for recognition of the -35 region: "Group V" sigma factors.

The results will be of broad interest to researchers in bacterial transcription and transcriptional regulation.

The structural figures have uninformative and inconsistent view orientations, excessive numbers of color, and non-standard and garish choices of colors. They will need to be redrawn before publication.

Specific comments:

59-63: Replace sentence with full paragraph citing and describing the previously published structures of Group I, II, III, and IV sigma-dependent transcription initiation complexes.

Figs: Use informative and consistent view orientations, avoid excessive numbers of color, and avoid garish choices of colors. Where possible, use literature-standard colors.

Fig. 1C: Show structures in consistent view orientations.

Fig. 2: Replace with figure showing (1) sigma-I region 4 (with view orientation, rendering,

and coloring that clearly shows HTH motif), (2) superimposition of sigma-I region 4 and sigma-A region 4 (with same view orientation, and with superimposition based on HTH motif), (3) sigma-I region 4 in complex with DNA (with same view orientation), and (5) superimposition of sigma-I region 4 in complex with DNA and sigma-A region 4 in complex with DNA (with same view orientation, and with superimposition based on HTH motif).

Fig. Compare the structure of sigma-I region 2 in complex with DNA to--in order--a Group I sigma region 2 in complex with DNA, a Group III sigma region 2 in complex with DNA, and a Group IV sigma region 2 in complex with DNA,

Fig 3B: Current images are uninformative (essentially piles of spaghetti). Replace with images having informative view orientations, carefully chosen rendering, and carefully chosen colors.

Reviewer #2 (Remarks to the Author):

In this manuscript, Li and his colleagues present the 3D structure of two transcription open complexes for two *C. thermocellum* σ factors, SigI1 and SigI6. The overall architectures of the complexes are similar to that of other known RPo complexes in various bacteria. Through a detailed structural analysis, the authors compare SigIC with other σ 4 factors and highlight its unique -35 element recognition mode, the presence of a conserved minor-groove A-tract in the recognition, and an approximately 180° rotation in the major-groove binding by HTH. The -10 element interaction is found to be more extensive but lacks specificity due to the non-conserved residues that bind to downstream bases of the -10 element. Furthermore, the authors show that the anti- σ factor RsgI inhibits σ I activity through competitive binding, as revealed by the presented structure.

This study provides insights into the mechanisms of bacterial transcription initiation, shedding light on some new binding modes in the σ I family's recognition of promoters. However, there are multiple major concerns that need to be addressed:

Major concerns:

(1) Generally, the σ family can be classified into four groups based on the presence or absence of four conserved helical structured domains (σ 1.1, σ 2, σ 3, σ 4). The authors proposed the classification of σ 1 as a new group V due to their unique DNA-binding mode, but this claim is overstated. Firstly, although SigIC binding with the -35 element exhibits a rotation and shift in structure analysis, it still functions the same way as other σ 4-domains, which serve as a contact point for sigma factors to bind -35 element of DNA promoter to stabilize holoenzyme and initiation complex. This indicates that SigI possesses a σ 4-like domain. Additionally, the authors indicate that SigI adopts a σ 1.2 region based on the position where helix α 1 is attached to the head of the oval structure. However, the binding residue for α 1 and the -10 element is non-conserved, and the binding mode is different from that of other σ 1.2 regions in groups I and II. Therefore, it's difficult to conclude that SigI possesses a canonical σ 1.2 region. Overall, SigI presents a σ 4-like and σ 2 domain, while lacking σ 1.1 and σ 3 domains. This suggests that it is better to subclassify SigI in group IV rather than classify it as a new group.

(2) In vivo experiments are easily affected by unknown factors. Authors should perform in vitro experiments to investigate and characterize the significance of the observed new sigma4-minor groove of -35 element, such as DNA binding analysis and in vitro transcription assay. In the reviewer's opinion, this interaction is not as important as the reported by the authors because it is pretty clear that main interaction between sigma4 and -35 element is located at the major groove of -35 element.

(3) Similarly, the effect of mutants should also be evaluated by in vitro experiments, such as EMSA and in vitro transcription assay.

(4) The local resolutions of sigma4 binding region in both RPo-SigI1 and RPo-SigI6 are all around 4.5 Å. This resolution is not good enough for unambiguously conducting side chain assignments. Additionally, figures with the density around this region are necessary for evaluating the quality.

Minor points:

(1) Fig. S5B SigI6 -8 nucleotide is A while all other data -8 nucleotides are shown as G. Please

correct it.

(2) The reported resolution of EMD-35131 is 3.36 Å, however, the provided FSC in pdb bank suggests it is 3.58 Å.

Reviewers' comments:

Reviewer #1 (Remarks to the Author):

The authors report cryo-EM structures of bacterial transcription initiation complexes dependent on the transcription initiation factor sigma-I. The structures show that sigma-I uses a novel DNA binding domain, different from those in other sigma factors, to recognize the promoter -35 region and reveal the architecture and interactions of the novel DNA binding domain.

The results indicate that sigma-I, which previously had been classified as a Group IV sigma factor, should be re-classified as the first member of a new class of sigma factors having the overall organization of Group IV sigma factors, but having a different structural module for recognition of the -35 region: "Group V" sigma factors.

The results will be of broad interest to researchers in bacterial transcription and transcriptional regulation.

The structural figures have uninformative and inconsistent view orientations, excessive numbers of color, and non-standard and garish choices of colors. They will need to be redrawn before publication.

Response: We carefully reconsidered the colour choices and figure styles. Most of the structural figures were redrawn in the revised manuscript in accordance with the reviewer's comments (see below), and we hope that the revised figures will better align with the reviewer's expectations.

Specific comments:

59-63: Replace sentence with full paragraph citing and describing the previously published structures of Group I, II, III, and IV sigma-dependent transcription initiation complexes.

Response: We have added a new paragraph for the description of the structures of each group.

Figs: Use informative and consistent view orientations, avoid excessive numbers of color, and avoid garish choices of colors. Where possible, use literature-standard colors.

Response: We have carefully reconsidered the orientations and colour choices in the revised manuscript.

Fig. 1C: Show structures in consistent view orientations.

Response: We revised this figure according to the reviewer's suggestion.

Fig. 2: Replace with figure showing (1) sigma-I region 4 (with view orientation, rendering, and coloring that clearly shows HTH motif), (2) superimposition of sigma-I region 4 and sigma-A region 4 (with same view orientation, and with superimposition based on HTH motif), (3) sigma-I region 4 in complex with DNA (with same view orientation), and (5) superimposition of sigma-I region 4 in complex with DNA and sigma-A region 4 in complex with DNA (with same view orientation, and with superimposition based on HTH motif).

Response: We have revised this figure according to the reviewer's comments.

Fig. Compare the structure of sigma-I region 2 in complex with DNA to--in order--a Group I sigma region 2 in complex with DNA, a Group III sigma region 2 in complex with DNA, and a Group IV sigma region 2 in complex with DNA,

Response: We revised this figure according to the reviewer's suggestion.

Fig 3B: Current images are uninformative (essentially piles of spaghetti). Replace with images having informative view orientations, carefully chosen rendering, and carefully chosen colors.

Response: The Figure (and Suppl. Fig.S8) was simplified to clarify the information presented, in line with the reviewer's comment.

Reviewer #2 (Remarks to the Author):

In this manuscript, Li and his colleagues present the 3D structure of two transcription open complexes for two *C. thermocellum* σ I factors, SigI1 and SigI6. The overall architectures of the complexes are similar to that of other known RPo complexes in various bacteria. Through a detailed structural analysis, the authors compare SigIC with other σ 4 factors and highlight its unique -35 element recognition mode, the presence of a conserved minor-groove A-tract in the recognition, and an approximately 180° rotation in the major-groove binding by HTH. The -10 element interaction is found to be more extensive but lacks specificity due to the non-conserved residues that bind to downstream bases of the -10 element. Furthermore, the authors show that the anti- σ factor RsgI inhibits σ I activity through competitive binding, as revealed by the presented structure.

This study provides insights into the mechanisms of bacterial transcription initiation, shedding light on some new binding modes in the σ I family's recognition of promoters. However, there are multiple major concerns that need to be addressed:

Major concerns:

(1) Generally, the σ family can be classified into four groups based on the presence or absence of four conserved helical structured domains (σ 1.1, σ 2, σ 3, σ 4). The authors proposed the classification of σ I as a new group V due to their unique DNA-binding mode, but this claim is overstated. Firstly, although SigIC binding with the -35 element exhibits a rotation and shift in structure analysis, it still functions the same way as other σ 4-domains, which serve as a contact point for sigma factors to bind -35 element of DNA promoter to stabilize holoenzyme and initiation complex. This indicates that SigI possesses a σ 4-like domain. Additionally, the authors indicate that SigI adopts a σ 1.2 region based on the position where helix α 1 is attached to the head of the oval structure. However, the binding residue for α 1 and the -10 element is non-conserved, and the binding mode is different from that of other σ 1.2 regions in groups I and II. Therefore, it's difficult to conclude that SigI possesses a canonical σ 1.2 region. Overall, SigI presents a σ 4-like and σ 2 domain, while lacking σ 1.1 and σ 3 domains. This suggests that it is better to subclassify SigI in group IV rather than classify it as a new group.

Response: Although we very much appreciate the reviewer's stance regarding the classification of

SigI, we continue to claim that SigI should belong to a new group (Group V) of the sigma-70 family for the following reasons:

(1) The initial classification of the sigma-70 family was based on sequence conservation of several regions, and later studies showed that this sequence conservation is largely in agreement with their structure/domain conservation (References 3 and 5 in the revised manuscript). From both the sequence conservation and the structural conservation, SigI cannot fit into any of the four groups, mainly because of its distinct C-terminal domain. Because groups I to IV have progressively less conserved regions/domains compared with the group I (SigA) factor and SigI has only one region/domain (σ_2) homologous to SigA – i.e., less than group IV whose members have two conserved regions – SigI should therefore be classified into a new group.

(2) We agree that SigIC binds the -35 element and this particular function is similar to that of other σ_4 domains, but we contend that this similarity is not the reason that it should be classified into group IV. Instead, the functional similarity could be the reason that it still belongs to the Sigma70 family, because all Sigma70 family factors bind to -10 and -35 elements.

(3) As shown in Figure 5A, the differences between the SigI and group IV factors are even greater than those between group III and group IV. Therefore, we believe that it is problematic to subclassify SigI in group IV.

The reviewer's comments here were very important for us and have prompted us to clarify to future readers of our article why SigI should be classified in its own group. Considering the above points, we have further revised the first paragraph of the discussion section to clearly state the reasons why SigI should be classified into a new group.

(2) In vivo experiments are easily affected by unknown factors. Authors should perform in vitro experiments to investigate and characterize the significance of the observed new sigma4-minor groove of -35 element, such as DNA binding analysis and in vitro transcription assay. In the reviewer's opinion, this interaction is not as important as the reported by the authors because it is pretty clear that main interaction between sigma4 and -35 element is located at the major groove of -35 element.

Response: We agree with the reviewer that in vivo experiments are possibly affected by unknown factors. Nevertheless, the heterologous Bacillus system has been used previously for study of the transcriptional activity of SigI and other sigma factors, in cases where either the homologous factor in Bacillus was knocked out or Bacillus lacked the paralogues of the heterologous sigma factor (References 27, 28, and 38 in the revised manuscript). Nevertheless, we performed an in vitro transcription assay for some of the mutants to address the question posed by the reviewer. The results are largely consistent with the in vivo experiments as shown in Fig.2E in the revised manuscript. These mutants contain not only the regions of the minor groove but also the major groove of the -35 element as well as the -10 element. Therefore, we maintain that the in vivo experiments are reliable and our conclusions are correct.

(3) Similarly, the effect of mutants should also be evaluated by in vitro experiments, such as EMSA and in vitro transcription assay.

Response: As explained in the latter response, we performed in vitro transcription assays and confirmed that the results are consistent with the in vivo findings. Although EMSA experiments are often used for detection of nucleic acid binding, it is seldomly used to reflect the transcriptional

activity of sigma factors. In view of the reviewer's suggestion, we have performed EMSA assays (shown in the figure below), and we found that we can observe the band shift only at very high SigI:DNA ratios (16 or 24), which raised the question whether the shifted band is a correct form of the SigI-RNAP-DNA complex. Further EMSA detection of several SigI mutants provided inconclusive results which cannot be explained in a reasonable manner. Therefore, we think that in this case the EMSA assay is not suitable for detection of the SigI-DNA interaction.

(4) The local resolutions of sigma4 binding region in both RPo-SigI1 and RPo-SigI6 are all around 4.5 Å. This resolution is not good enough for unambiguously conducting side chain assignments. Additionally, figures with the density around this region are necessary for evaluating the quality.

Response: Indeed, the local resolutions of sigma4 binding region are not as high as those of the RNAP core region. At this resolution, we cannot observe the side chains for all of the residues. This is the reason why we could not analyze and discuss much about the detailed interactions (such as hydrogen bonding pattern). However, with the help of the AlphaFold structural model and our previously determined structures of SigI1C and SigI6C complexed with RsgI, we found that the structural model fits well into the density, and some large side chains (particularly aromatic side chains) can be observed with appropriate density. Therefore, we believe our structural models are largely reliable when we discuss the residue-level interactions. We added a new Suppl. Fig S4 to show the densities of the sigma4 domain and -35 element DNAs. We also explained the resolution problem relating to this region, in the section “Interactions between σ I and promoter -35 element” of the revised manuscript.

Minor points:

(1) Fig. S5B SigI6 -8 nucleotide is A while all other data -8 nucleotides are shown as G. Please correct it.

Response: We thank the reviewer and have corrected this mistake in the revised manuscript.

(2) The reported resolution of EMD-35131 is 3.36 Å, however, the provided FSC in pdb bank suggests it is 3.58 Å.

Response: This difference is because we used different methods for RPo-SigI1 and RPo-SigI6 in the final map refinement. Unlike the RPo-SigI1 dataset, the density of the SigI6-DNA binding region appeared blurred and fragmented, indicating greater flexibility in this region. Consequently, for the RPo-SigI6 dataset, a deep learning-based approach, CryoDRGN, was used to classify particles, and three similar classes were selected to perform the non-uniform refinement. The PHENIX Density modification process was then used to improve the phase and the resolution (Terwilliger, T.C. et al. Nat Methods 2020, 17, 923–927). The reason that we employed this approach was to improve the accuracy of the model building. Using this approach, we did not calculate the resolution using the traditional half map gold standard of 0.143 as used in the PDB Bank but used map-to-perfected model calculations to estimate FSC (Dubach, V.R.A. et al. Crystals 2020, 10, 580.). This approach has also been applied in other studies when authors sharpened their final map (Kern D.M. et al. Nat. Struct. Mol. Biol. 2021, 28:573-582; Noviello, C. M. et al. Cell 2021, 184:2121-2134.e2113; Nadezhdin, K. D. et al. Nat Struct Mol Biol 2021, 28:564-572). Hence, the reported resolution for EMD-35131 is 3.36 Å (using density modification with an FSC cutoff at 0.5), while the provided FSC in the PDB bank suggests it is 3.58 Å (using raw half map indicated at a 0.143 cutoff) for RPo-SigI6. This has been described in the method of data processing part of the RPo-SigI6 data set. In addition, to clearly describe the differences in the map refinement of the two complexes, i.e., RPo-SigI1 and RPo-SigI6, we have also revised the related part in the “Cryo-EM data acquisition and processing” section of Methods.

REVIEWER COMMENTS

Reviewer #1 (Remarks to the Author):

The revisions to figures have improved the manuscript.

The manuscript now should be acceptable for publication, but could be improved by using less garish colors for RNAP beta and beta' subunits in Figure 1.

Reviewer #2 (Remarks to the Author):

The authors have provided additional in vitro transcription data and tried EMSA analyses (not successful). Some relative concerns have been addressed, however, it's still hard for this reviewer to agree with the current version.

(1) Just as this reviewer has stated in last report, this sigma factor (sigma1) has two identified functional domains for promoter recognition: the conserved sigma2 for -10 element recognition, and sigma1C (a HTH motif + coil-coil) for recognizing -35 element, although sigma1C is different from sigma4 of sigmaA in the sequence and structure to a certain extent. However, the composition of domains for recognizing DNA promoter is the key factor for classifying sigma factors: Group I (sigma70/sigmaA -example): $\sigma_{1.1}$, σ_2 ($\sigma_{1.2} + \text{NCR} + \sigma_2$), σ_3 , and σ_4 ; Group II (sigma38/sigmaS -example): σ_2 ($\sigma_{1.2} + \sigma_2$), σ_3 , and σ_4 ; Group III (sigma28 -example): σ_2 (σ_2), σ_3 , and σ_4 ; Group IV (ECF sigma factor -example): σ_2 (σ_2) and σ_4 . It's clear that Group IV of sigma 70 family has two identified domains as well: sigma2 (for -10) and sigma4 (for -35). Thus, sigma1 is certainly a member of "group IV", not the one of a new group, although it may be an unusual Group IV member.

Other comments:

(2) In addition, please correct Figure 5A:

NCR should not be shown in Group II (the representative sigma38 in Group II doesn't have a typical NCR); $\sigma_{1.2}$ should not be shown in Sigma1 because authors claimed a $\sigma_{1.2}$ region based on the position where helix α_1 is attached to the head of the oval structure, however, the binding residues for α_1 and the -10 element is non-conserved, and the binding mode is different from that of other $\sigma_{1.2}$ regions in groups I and II, which has been stated in the last

report.

(3) In the abstract, it is not suitable to claim, “the binding of which differs completely from that of other σ_{70} -family 33 sigma factors that bind to the major groove alone, with a 180° rotation compared to the σ_{1} factors”. In light of Figure 2A, it’s clear that the HTH motif is the main force to recognize the major groove at the -35 element for both σ_{1} and σ_{4} of σ^A . Additionally, it’s hard to get the conclusion of 180° rotation based on the available figures.

Reviewers' comments:

Reviewer #1 (Remarks to the Author):

The revisions to figures have improved the manuscript.

The manuscript now should be acceptable for publication, but could be improved by using less garish colors for RNAP beta and beta' subunits in Figure 1.

RESPONSE: We thank the reviewer for the positive comment and suggestion on the Figure 1. We changed the colors of RNAP beta and beta' subunits to green and cyan in the revised Figure 1, respectively.

Reviewer #2 (Remarks to the Author):

The authors have provided additional in vitro transcription data and tried EMSA analyses (not successful). Some relative concerns have been addressed, however, it's still hard for this reviewer to agree with the current version.

(1) Just as this reviewer has stated in last report, this sigma factor (sigmaI) has two identified functional domains for promoter recognition: the conserved sigma2 for -10 element recognition, and sigmaIC (a HTH motif + coil-coil) for recognizing -35 element, although sigmaIC is different from sigma4 of sigmaA in the sequence and structure to a certain extent. However, the composition of domains for recognizing DNA promoter is the key factor for classifying sigma factors: Group I (sigma70/sigmaA -example): $\sigma_{1.1}$, σ_2 ($\sigma_{1.2+}$ NCR+ σ_2), σ_3 , and σ_4 ; Group II (sigma38/sigmaS -example): σ_2 ($\sigma_{1.2+}$ σ_2), σ_3 , and σ_4 ; Group III (sigma28 -example): σ_2 (σ_2), σ_3 , and σ_4 ; Group IV (ECF sigma factor -example): σ_2 (σ_2) and σ_4 . It's clear that Group IV of sigma 70 family has two identified domains as well: sigma2 (for -10) and sigma4 (for -35). Thus, sigmaI is certainly a member of "group IV", not the one of a new group, although it may be an unusual Group IV member.

RESPONSE: We thank the reviewer for the detailed comments on the classification. We revised the manuscript and no longer classify SigI into a new group.

Other comments:

(2) In addition, please correct Figure 5A:

NCR should not be shown in Group II (the representative sigma38 in Group II doesn't have a typical NCR); $\sigma_{1.2}$ should not be shown in SigmaI because authors claimed a $\sigma_{1.2}$ region based on the position where helix α_1 is attached to the head of the oval structure, however, the binding residues for α_1 and the -10 element is non-conserved, and the binding mode is different from that of other $\sigma_{1.2}$ regions in groups I and II, which has been stated in the last report.

RESPONSE: We thank the reviewer for pointing it out. We removed NCR in Group II and denote the helix α_1 of SigI as $\sigma_{1.2}$ -like in the revised manuscript.

(3) In the abstract, it is not suitable to claim, "the binding of which differs completely

from that of other σ 70-family 33 sigma factors that bind to the major groove alone, with a 180° rotation compared to the σ I factors". In light of Figure 2A, it's clear that the HTH motif is the main force to recognize the major groove at the -35 element for both σ I and σ 4 of σ A. Additionally, it's hard to get the conclusion of 180° rotation based on the available figures.

RESPONSE: We thank the reviewer for the comment and sorry that we may have not described the 180° rotation clearly in our previous manuscript. The roles of major and minor groove binding were validated by mutation of key residues reported in our manuscript and mutation of nucleotides reported in a previous study (References 27 and 28). Therefore, the conclusion about the minor groove is not derived from a simple observation of the structure. For the 180° rotation, we labeled the 5'-3' direction of the NT strands in the right-hand panel of the previous Figure 2A, and it clearly shows that the DNAs exhibit a 180° rotation (i.e., they run in the opposite direction) when the structures are aligned according to the helix-turn-helix motif. In the new version of Figure 2A, we added a new panel in which the structures are aligned according to the DNAs, and the 180° rotation of the HTH motifs in the two structures can be seen clearly. We also revised the figure legends to indicate this point. We hope the changes on the revised figure and text could address this issue with satisfactory.

REVIEWERS' COMMENTS

Reviewer #2 (Remarks to the Author):

I appreciate the efforts that the authors have made in response to my concerns. The current version clarifies all the points I raised. I support it to be published.